# Polymerized β-Cyclodextrin-Based Injectable Hydrogel for Sustained Release of 5-Fluorouracil/Methotrexate Mixture in Breast Cancer Management: In Vitro and In Vivo Analytical Validations

**DOI:** 10.3390/pharmaceutics14040817

**Published:** 2022-04-08

**Authors:** Saud Almawash, Mohamed A. El Hamd, Shaaban K. Osman

**Affiliations:** 1Department of Pharmaceutical Sciences, College of Pharmacy, Shaqra University, Shaqraa 11961, Saudi Arabia; salmawash@su.edu.sa (S.A.); aboelhamdmohamed@su.edu.sa or; 2Department of Pharmaceutical Analytical Chemistry, Faculty of Pharmacy, South Valley University, Qena 83523, Egypt; 3Department of Pharmaceutics and Pharmaceutical Technology, Faculty of Pharmacy, Al-Azhar University, Assiut 71524, Egypt

**Keywords:** cell viability, tumor markers, cholesterol, branched PEG, self-assembling hydrogel, controlled release, pharmacokinetics

## Abstract

An inclusion complexation, between polymerized β-cyclodextrin and cholesterol end-capping branched polyethylene glycol, was utilized for constructing a self-assembled hydrogel. The physicochemical properties, the in vitro release profiles of 5-Fluorouracil/methotrexate (anticancer drugs), and the surface morphology of the resulting hydrogel were studied. Moreover, in vivo studies were carried out on female rats bearing breast cancer. The results revealed that the prepared systems were white in color, rubbery, and homogenous. The in vitro release studies showed an efficient ability of the modified system for drug loading and release in a sustained release manner for 14 days. The surface morphology was spongy porous. Moreover, the tumors’ healing was indicated from the analysis of tumor volume, plasma tumor markers, and histopathological analysis, compared to the controlled rats. The pharmacokinetic parameters appeared significant differences (*p* < 0.05) in the C_max_ and T_max_ of the medicated hydrogel samples, as compared with sole or combined saline-injected samples. The whole AUC of each drug in the medicated hydrogel samples was five-fold more than the mixture administrated in PBS. In conclusion, the proposed work delivered a hydrogel system that has a convenient ability for localized sustained release of breast cancer management.

## 1. Introduction

Cancer has been a significant impediment to rising life expectancy and a leading cause of death [1]. An indicator of cancer spreading was confirmed by the global survey, with 18 million cases in 2018 of different cancerous diseases; of those, there were 8.5 million cases with women [2]. However, breast cancer is the most common cause of female cases [3]. No less than 90% of breast cancers are not metastatic at the time of cancer discovery and diagnosis. Therefore, for these patients, the therapeutic goals are tumors’ eradication and preventing their recurrence via surgery as a rapid treatment, plus pre- or postoperative systemic chemotherapy.

A detail for the strategies of local and systemic therapies that are used for breast cancer was specifically summarized in a recently published review [4]. However, the chemotherapy regimens remain an essential treatment for preventing the tumors’ recurrence with all different stages of breast cancer [5]. Both single and combination chemotherapy is used, depending on the severity of the disease. Eventually, the combination regimens demonstrated a further improvement in the sound of the clinical response and progression. The timeline of chemotherapies was initiated in 1976, with cyclophosphamide, 5-fluorouracil (5-FU), and methotrexate (MTX) [5]. However, the clinical success of 5-FU is limited, owing to its improper pharmacokinetics profile, such as low bioavailability, inadequate selectivity, high cell toxicity, short half-life (10–20 min), and the appearance of drug resistance [5,6,7]. MTX is the drug of choice for potentiation of 5-FU uptake by cancerous cells, energizing its effect via inhibition of purine biosynthesis, and decreasing its undesirable side effects [3]. Unfortunately, the oral or parental administration of both drugs may be caused a fast-uncontrolled release and instant high peak of 5-FU, followed by a sharp decline of plasma level, which is collectively responsible for the non-selectively exposure to healthy and noncancerous tissues, leading to an additive and severe systemic toxicity [8,9].

On the other hand, the intratumorally injection may be one of the techniques that cause an efficient increase in the therapeutic concentration level of 5-FU and MTX mixture inside the intended tumors and, consequently, minimizes the systemic drug exposure to healthy and non-cancerous tissues [10]. Recently, an injectable medicated hydrogel is considered an intratumorally injection for targeting and localization some drugs at the tumor site, sustaining their release and action, as well as being expected to improve the patient compliance and avoid the surgical interventions, as required in such cases of implants [3,4]. Hydrogels are considered as the best targeting models for high drug localization at the tumor site, sustained drug action, improved patient compliance, and avoiding surgical interventions as required in the case of implants [3,11]. Beta Cyclodextrins (β-CD), a cyclic cone-shaped seven-glucose structure of hydrophilic exterior and hydrophobic interior, can form inclusion complexes with different hydrophobic molecules [12]. This property was utilized previously, to form self-assembling hydrogel networks [13,14,15]. However, native β-CD cannot form stable self-assembling hydrogels as their poor solubilities and low molecular weights (18 mg/mL) [14]. Therefore, the polymerization of β-CD could overcome such a problem, since it was transformed from crystalline to an amorphous one of higher solubility [16]. Recently, stable supramolecular hydrogels were formed from the hydrophobic interactions between CD-derived polymers and hydrophobic guest molecule end-capping polymers, such as hydrophobically modified dextran [13,14,15], hydrophobically modified polyacrylamide copolymer [17], and high-molecular-weight polyethylene glycols (PEG)s [18], which are used in controlled drug delivery applications.

Although hydrogels based on the inclusion complexation between pβ-CD and cholesterol end-capping polyethylene glycol were reported previously [19], their full potential as an anticancer drug delivery was not revealed yet. Therefore, we focused, in the present work, on the in vitro and in vivo application of the modified hydrogel, for the first time, in breast cancer management via a local intratumorally injection of combination of two anticancer drugs (5-FU and MTX). Consequently, the modified hydrogel gives the chance for controlled release of the chemotherapeutic agent, which decreases the dangerous side effects accompanying the fluctuation in the drug concentration, which caused by repeated administration of chemotherapeutic drugs. We, therefore, developed a breast tumor model in rats to study the in vivo action of 5-FU/MTX loaded on the evolution of the breast tumor, in comparison with the free aqueous saline solutions of both 5-FU and MTX. Furthermore, we tried to better understand the toxicological influences and to investigate the cellular inflammatory infiltrate, which surround the tumor areas through the investigation of histopathological traits and tumor markers (CEA, CA, TNF-alpha, and Interleukin-6 (IL-6)), which were measured in the blood samples of rats before and after treatment to evaluate the suppression effect on tumor cells by the modified hydrogels. In addition, the pharmacokinetic parameters (Ka, Ke, T_1/2,_ AUC, C_max_, and T_max_) were investigated by HPLC and reported in the present work for better understanding the kinetics of the drugs inside the body, which have never investigated before. Furthermore, we paid attention to the cytotoxicity of the modified hydrogel (unmedicated and medicated), which was investigated using the MTT assay with the MCF-7 breast cancerous cell membranes.

## 2. Materials and Methods

### 2.1. Materials

The 5-fluorouracil (5-FU) was purchased from Applichem for Pharmaceuticals Co., GmbH (Darmstadt, Germany). Methotrexate (MTX) was obtained from Haupt Pharma Wolfratshausen GmbH (Chemajet Co., (Wolfratshausen, Bayern, Germany). β-Cyclodextrin (β-CD) and epichlorohydrin (EP) were obtained from Wacker Fine Chemicals (Wacker Chemie AG, Burghausen, Germany). Diisopropyl azodicarboxylate (DIAD), brine, Toluene, and succinic anhydride (SA) were delivered from Sigma-Aldrich (Schnelldorf, Germany). Cholesterol, triphenylphosphine (PPh3), phthalimide, dibutyltin dilaurate (DBDL), and dimethyl aminopropyl-ethyl carbodiimide (EDC) were obtained from Acros Organics (Geel, Belgium). Linear eight-armed poly (ethylene glycol) with a molecular weight of 20 kDa (8armPEG20k-OH) was delivered from Jenkem Technology (Beijing, China). Ethanol, tetrahydrofuran (THF), triethylamine (TEA) dichloromethane (DCM), and the other chemicals of an analytical grade were obtained from Merck KGaA (Darmstadt, Germany). Phosphate buffered saline (PBS), deuterated water, and chloroform (respectively, d6-D_2_O and d_6_-CDCl_3_) were purchased from Deutero (Kastellaun, Germany). Celite 503 were obtained from Fluka (Taufkirchen, Germany).

### 2.2. Synthesis of the Hydrogel Base

#### 2.2.1. Synthesis of the Host Molecules

A native anhydrous β-CD was polymerized according to the previously reported method [17], into a soluble pβ-CD using EP as a crosslinking agent in an alkaline medium. Briefly, a native β-CD (10 g, 8.82 mmol) has been dissolved in 15 mL NaOH solution (15% *w*/*v*) and allowed to be stirred at 35 °C for 2 h. Toluene (2 mL) was added to the above alkaline solution and was stirred at the same temperature for a further 2 h. After, a calculated amount of EP (4.62 g, 50 mmol) [17] was added to that mixture and allowed to be stirred for 3 h. Then, the crude product was collected via precipitation of the solution mixture in 200 mL isopropanol solvent. Thus, the formed polymerized precipitate was dissolved in distilled water and neutralized to pH 7 by a dropwise addition of a diluted HCl (1 M). To remove the small molecular weight polymers and other impurities or excess of toxic organic solvents, the solution of the formed polymer was dialyzed for one week against distilled water (N.B., the distilled water was changed twice daily to accelerate the dialysis process), using a dialysis sac of MWCO 14 kDa. Finally, the pure product was collected by lyophilization. The obtained product was analyzed for degree of substitution and molecular weight by ^1^HNMR, ^13^C NMR spectroscopy, and the Dynamic light scattering (DLS) technique.

#### 2.2.2. Synthesis of the Guest Part

Osman et al. [19] has developed a coupling method for the preparation of branched 8-arm polyethylene glycol-cholesterol (8-armPEG20k-chol), after the formation of an amide bond with the aid of a previously prepared molecule of cholesteryl-PEG. Through the steps of the present synthesis, an amount of a purified 8armPEG20k-amines (1 mmol), synthesized according to the previously reported procedure [17], was dissolved together with cholesteryl-oxypropionic acid (2 mmol/each amino group) in dry dichloromethane under nitrogen gas. Following a cooling down of the solution till 0 °C, both EDC and HOBT were used as the coupling reagents (1.5 mmol/each amino group). After adding an amount of TEA solution (1.5 mmol/each amino group); the whole mixture was allowed to be stirred under nitrogen gas at 0 °C for 4h. Then, the stirring was continued at 25 °C for a further 72 h. Next, an excess of an organic solvent DCM was added to dilute the mixture, and, then, sequentially, saturated solution of ammonium chloride (5 mL), HCl solution (5 mL, 5 M), and brine (5 mL) were added. Finally, the product was collected via precipitation in ether. Then, for further purification, the residue was dissolved in DCM and reprecipitated in diethyl ether three times. The synthesized polymer was analyzed by NMR, MALDI, and GPC for purity degree and molecular weight determination.

### 2.3. Construction of the Hydrogel Base Loaded with 5-FU/MTX Mixture

Initially, a solution of both pβ-CD polymer and branched 8-armPEG-chol (10%, *w*/*v* in distilled water of 1:1 ratio) was freeze-dried to obtain a fluffy powder of polymer mixture. Subsequently, the hydrogel was constructed via the rehydration of the obtained fluffy powder by PBS. Noteworthy, the lyophilization of the mixture could help in enhancing the solubility of cholesterol-PEG [19]. The drug loading can be performed during the rehydrating process using PBS containing 5-FU/MTX (0.1%, *w*/*v*, in a rehydrating solution), which was added and distributed well.

The formulated hydrogel was visually detected for their color, the existence of 3D structure, transparency, smoothness, and homogeneity [20]. Then, the pH (measured by using a pH meter) and drug content uniformity were investigated.

Regarding drug content measurements, each sample of medicated hydrogel (1.0 g/mL, containing 0.1% of each drug in the mixture) was mixed in a vortex in an excess of distilled water, to destroy the gel network, and then the contents of the drug mixture were collected after filtration. Then, 1.0 mL of the filtrate was taken and measured by a simultaneous spectrophotometric method, developed, and validated by the authors, at specific maximum wavelengths, of λ_max_ 266 nm and 302 nm for 5-FU and MTX, respectively [21,22].

### 2.4. Rheological Studies

The rheological behavior of the medicated hydrogel system was explored, using a cone and 20 mm diameter of a plate AR 2000 rheometer (TA Instruments, Eschborn, Germany), attached with a thermostatic water bath. After, the upper plate lowered gradually to touch the sample mass, regarding the previously specified gap size (1 mm). The viscoelastic behaviors of a number of the medicated hydrogel samples (n = 3) were evaluated by measuring two important parameters named G′ (storage modulus) and G″ (loss modulus) as a function of shear stress (μN·m). Generally, when the value of G′ in each hydrogel sample is higher than that of G″, the system can be identified as a true ward viscoelastic hydrogel. Otherwise, it will be considered as a viscous liquid if the sample value of G″ is higher [23]. Additionally, the hydrogel strength of different samples (n = 3) and their stability were investigated via recording both the complex shear modulus (G*) and the torque, at which the tested hydrogel samples will be destroyed, when the fixed parameters (1Hz frequency, 10 (μN·m), and 25 °C) are adjusted. Finally, the irreversibility of the medicated hydrogel samples was confirmed after measuring their hydrogel temperature (T_gel_), after transitioning the network of their hydrogel into a solution, with the aid of an oscillatory temperature sweep experiment.

### 2.5. In Vitro Release Studies of the Medicated Hydrogel Samples

Several samples (n = 6) of the medicated hydrogel (0.5 g, containing 0.5 mg of each 5-FU and MTX), were placed in 2-mL glass vials and allowed to be incubated at 4 °C for 2 h. Next, one mL of PBS was added above the formulated hydrogel mass and allowed to be incubated in a thermostatic water bath under a shaking rate of 50 rpm at 37 °C. After specific time intervals (each 24 h), 700 µL of the supernatant was withdrawn and replaced immediately with a fresh-prepared PBS, previously incubated at the same temperature to ensure the sink conditions [24]. This process of sample withdrawal and replacing was allowed to continue until the complete clearance or full release of both two incorporated drugs from the hydrogel samples. Thus, the concentration of 5-FU and MTX in the mixture, in each collected sample, was measured simultaneously at 266 nm and 302 nm, respectively, with the aid of the spectrophotometric method, as mentioned before [22,23,24,25].

### 2.6. Syringeability and Injectability Study

The injectable hydrogel system should have a suitable consistency, which can be extruded properly through the syringe needle into skin layers for subcutaneous injection [26]. The injectability studies were performed by filling a syringe-needle (5 mm needle size) with either 5-FU solution of 5-FU loaded hydrogel and injecting them into a meat sample under the finger pressure.

### 2.7. In Vitro Antitumor Activity and Cell Viability

The in vitro antitumor activities of the prepared hydrogel system were evaluated against breast cancer cells (MCF-7) by 3-(4,5-dimethylthiazol-2-yl)-2,5-diphenyl-2H-tetrazolium MTT-assay [27]. MCF-7 were cultivated in Dulbecco’s Modified Eagle Medium (DMEM) (BioSera, Rue de la Caille, Muaille, France), supplemented with 10% *w*/*v* heat-inactivated fetal bovine serum (FBS) (BioSera, France) and 1% *w*/*v* antibiotics (100 I.U/mL penicillin and 100 µg/mL streptomycin) (PAA Laboratories GmbH, Austria). Firstly, cells were adjusted to 1 × 10^6^ cells per ml and seeded into a 96-well plate with 100 μL in each well and incubated for 24 h at 37 °C. Then, the culture medium was replaced with a volume of 100 μL of sterile unmedicated hydrogel system (control), 5-FU^®^ saline solution, 5-FU/MTX saline solution mixture, and a hydrogel system loaded with (5-FU/MTX) at different concentrations (100, 200, 400, and 600 μg/mL), and, then, they were incubated at 37 °C. Noteworthy, the investigated samples were previously sterilized by filtration method using microfilters. Thereafter, 100 μL of fresh culture medium was added to each well, and then cultured at 37 °C for 24 h. After incubation, the cells were treated with 25 μL of the MTT reagent at a concentration of 5 mg/mL in PBS and incubated for a further 4 h. Consequently, the media was carefully discarded, and 200 μL DMSO was added to solubilize the formazan crystals [28]. After 5 min of shaking, the optical densities were determined at 570 nm using an ELISA microplate reader (Bio-Tek, Winooski, VT, USA) [29]. All experiments were performed in triplicate.

### 2.8. In Vivo Studies

#### 2.8.1. Animal Preparation

The present in vivo works was performed on albino female rats (weighing 185 ± 15.1 g). The ethical approval of all the experiments was issued by the Faculty of Pharmacy, Al-Azhar University, under no AZ-AS/PH/2/C/2021. The rats were acclimatized at room temperature with a 12-h light/12-h dark cycle [30,31]. Tumor-induction of 48 female rats were randomly divided into the four independent groups (with n = 12 per each group × triple trials); G1 was the controlled group, which is positive control and were received no any type of medications; G2 received a sole dose of 5-FU saline solution (100 mg/kg); G3 received a combined mixture of 5-FU/MTX saline solution (respectively, 100 and 40 mg/kg); and, finally, G4 received 5-FU/MTX hydrogel. The sole 5-FU and/or combined with MTX doses in normal saline were injected into rats using normal syringes, while the injectable medicated hydrogel doses were injected using a mixing syringe device (Doowon Meditec Corp., Youngin City, Republic of Korea). After finishing the administration processes, blood samples were collected at certain time intervals (0.5, 1, 2, 4, 6, 8, 10, 12, and 14 h). HPLC chromatographic analysis was utilized for the detection of drug concentrations.

#### 2.8.2. Plasma Samples Pretreatment

One mL of a plasma sample in an Eppendorf tube was initially mixed with phosphate buffer (0.2 mL of 0.5 M at pH 8), after being mixed in a vortex, and an ethyl acetate solution (4.0 mL) was added and mixed in a vortex for 1 min. Then, the contents of the tube were centrifuged at 5000 rpm and temperature 10 °C for 5 min. The organic phase was taken and evaporated until dry, by using a stream of nitrogen gas at 55 °C. The drug mixture in the residue was reconstituted by a solution of potassium dihydrogen phosphate (100 µL of 0.01 M, at pH 4.0), and, then, the resulted solution was filtrated through a 0.2 μm filter syringe, before transferring into autosampler vials of the chromatographic analysis.

#### 2.8.3. Chromatographic Analysis of the Drugs’ Mixture in Plasma

A high-performance liquid chromatography, HPLC (Parkin Elmer, JASCO Corporation, Tokyo, Japan), equipped with an autosampler, was used through this study. The detection method was performed using a variable wavelength UV detector, which adjusted at 266 nm [32] for both drugs. The analytical Phenomenex, C18 symmetry RPh-column (4.6 × 250 mm; 5 µm PS) (Cap cell Pak UG120, Shiseido, Tokyo, Japan), was used at 40 °C. The mobile phase was an isocratic mixture (89:11, *v*/*v*) of, respectively, methanol and potassium dihydrogen phosphate (10 mM and adjusted to pH 4 with the aid orthophosphoric acid), which was pumped at a flow rate of 1.0 mL/min. The injected volume was 20 µL.

#### 2.8.4. Pharmacokinetic Study

The concentrations of 5-FU and MTX in different rats’ plasma samples (n = 6) were calculated from the corresponding calibration curves equation, developed in the above HPLC method. The plasma concentration-time (Cp/) curve of 5FU/MTX (plasma concentrations; µg/mL in case of using a vehicle of saline, and ng/mL in case of using a vehicle hydrogel) was constructed for both drugs. Next, from the constructed (Cp/t) curves, all the common pharmacokinetic parameters of both drugs were determined using the following Equation (1) [33].
(1)$$\left(\mathrm{Cp}\right)t=\frac{F\;\mathrm{Ka}\;X_{0}}{\mathrm{Vd}*\left(\mathrm{Ka}-\mathrm{Kel}\right)}\times (e^{-\mathrm{Kt}}-e^{-\mathrm{Kat}})$$
where Cp is the plasma concentration; Ka is the absorption rate constants, F is the absorption fraction of the drug bioavailability; X_0_ is the drug dose; and Vd is the volume of distribution and Ke is the elimination rate constant. Moreover, the area under the Cp/t curve (AUC_0-24_ and AUC_0-∞_) (ng. h/mL) was measured using the trapezoidal rule to indicate the extent of drug absorption. The peak Cp (C_max_), the time corresponding to the maximum Cp (T_max_), and the half-life, t_1/2_ (h), were calculated [34].

### 2.9. Histopathological Appraisals and Traits

Several tissue samples (n = 3) were taken from the tumor masses of the induced mammary glands and were fixed in formalin for 24 h. The fixed samples were processed for the usual histological procedures for hematoxylin and eosin stains, the counter staining of ultra-thin paraffin sections (5 µm; Micro HM 360^®^ Microtome, Marshall Scientific, Hampton Virginia, USA) [35]. The stained sections were examined under a light microscope (Olympus B × 46) and were digitally photographed using its connected Olympus DP 21 digital camera (Olympus Corporation, Tokyo, Japan).

### 2.10. Effect of the Proposed Hydrogel System on the Tumor Growth

For transplantation experiments, recipient syngeneic female rats (6–7 weeks old) were anesthetized with an intramuscular injection of ketamine (8.0 mg/kg dose) and xylazine (10 mg/kg dose). Breast cancer was induced to the investigated rats by chemical induction using DMPA, previously reported techniques [36,37]. Tumors ~1 cm in diameter were noted in all rats two weeks after tumor induction. The tumor size for all animals was examined using a Vernier caliper (micrometer-Ozaki Ltd., Tokyo, Japan) and calculated using the following Equation (2) [32].
TV (mm^3^) = 1/2 × Length × Width^2^(2)

Thus, the changes in a tumor volume in rats were determined every 5 days for 15 days after administration of the injectable medicated hydrogel (contains a concentration of 100 and 40 mg/kg, respectively, for 5-FU and MTX). The antitumor effect was monitored via measuring both the relative tumor volume (RTV), representing the tumor volume after treatment, using Equation (3) [25,38].
(3)$$\%\;\mathrm{RTV}=\left[\frac{\left(V_{t}\right)}{\left(V_{0}\right)}\right]\times 100$$
where, Vt represents the tumor volume at each time interval, and V0 is a tumor volume directly after drug injection. The % of tumor volume inhibition was calculated using the following Equation (4) [37].
(4)$$\%\;\mathrm{Inhibition}\;=\left[1-\frac{\left(V_{t}\right)}{\left(V_{0}\right)}\right]\times 100$$

### 2.11. Detection of the Tumor Markers in Rats’ Blood

To investigate the existence and the progress of the tumor, a blood sample was taken from the animals and subjected to tumor markers investigation. Two inflammatory mediators Tumor Necrosis Factor-alpha (TNF-α) and Interleukin-6 (IL-6) were evaluated as the tumor signals in the present study. Enzyme-linked Immunosorbent Assay Kit (Anogen, a division of Yes Biotech Laboratories Limited, Ontario, Canada) was utilized in the detection of these tumor markers [38]. The normal ranges of IL-6, TNF-α, and CRP in umbilical cord blood for healthy term Trinidadian neonates are 0–16.4 pg/mL, 0–29.4 pg/mL, and 0–12.4 mg/L, respectively [38].

### 2.12. Statistical Analysis

The statistical analysis was conducted on the normally distributed data, using the Statistical Package for the Social Sciences (SPSS) version 25 (IBM Corporation, Armonk, NY, USA) software [34]. Data were calculated from three independent experiments and listed as the mean values ± SD. Covariance analysis (ANOVA) was performed for comparison between the more than two samples investigated, and the differences between them were tested and checked for their statistical significance with the Duncan post hoc test used for the paired comparison of means [39]. Differences between data sets were considered statistically significant at *p* < 0.05, with more than 95% confidence level, while the *p*-values of <0.01 were considered highly significant in some cases of analysis.

## 3. Results and Discussion

### 3.1. Synthesis and Characterizations of the Hydrogel Base

pβ-CD was synthesized via polymerization of β-CD in a strong alkaline medium using EP as cross-linking agent. The conversion extent (DS) of native cyclodextrin into pβ-CD was 62%, as indicated from ^1^H NMR and ^13^C NMR integrations using the CD anomeric proton (7H for each CD molecule) at 5.1 ppm as a reference peak. The protons of EP appeared at 2.6–2.9 ppm (CD-O-CH_2_CH(OH)CH_2_-O-CD). In addition, the molecular weight was determined according to previously reported technique [17,18], since they reported that there is a relationship between the molecular weight and the hydrodynamic diameter of the formed polymer in solution. Noteworthy, they listed the effect of EP concentration on hydrodynamic diameter and molecular weight. Accordingly, our hydrodynamic diameter was 8.2 nm, which matches the molecular weight of 106 kDa. The results, which were presented in Table 1 and in the Appendix A) showed a resonance signal that matched what was obtained previously [17,18].

On the other hand, the guest molecule, 8armPEG20k-chol, was synthesized by coupling the 8armPEG20k-amines with cholesterol succinate in an anhydrous DCM and in the presence of HOBT and EDC as coupling agents under an N_2_ condition at 0 °C. The results showed that the synthesized 8armPEG20k-chol was obtained in a yield of 85%, with a degree of substitution of 87%, as indicated by ^1^H NMR, MALDI, and GPC. The 8armPEG20k-chol was obtained in a yield of 85%, with a degree of substitution of 87%, as illustrated by the ^1^H NMR (CDCl_3_, 600 MHz). NMR was utilized, also, for determination of the molecular weight of the formed polymer, considering the degree of substitution. The degree of substitution is defined as the number of cholesterol units per PEG molecule, which was calculated by comparing the relative peak area of the cholesterol moieties olefinic protons (at 4.6–5.4 ppm) to that of the PEG proton (at 3.2–3.9 ppm). Especially, in our case of 8arm PEG20-(Chol)_8_, each molecule of star-shaped PEG20k was planned to be decorated with 8 molecules of cholesterol. The molecular weight of each 8armPEG and cholesterol molecule is 20,000 Da and 386 Da, respectively. Therefore, the degree of substitution can be calculated from the number of cholesteryl-protons per each PEG molecule. Moreover, the molecular weight of the final product can be calculated so easily from the DS. For example, if the PEG20K was decorated with 7 cholesterol units (as in our case), the Mwt can be calculated as follows: Mw of 8armPEG20k-(chol)_7_ = (20,000 + (7*386)) = 22,702 Da [19].

### 3.2. Assembling the Hydrogel Base and Loading It with 5-FU/MTX Mixture

The hydrogel system was successfully constructed, by simply mixing the synthesized guest molecule’s solution with the host molecule solution. The trial experiments (n = 3) confirmed that the construction of the proposal hydrogel system was based on an inclusion complexation between the high molecular weight pβ-CD polymer and a cholesterol moiety in the guest polymer. Regarding the drug-mixture loading, the calculated amounts of both 5-FU and MTX were initially incorporated into the rehydrating solutions (PBS), to increase the drug’s affinity to the hydrogel base and consequently gave a distributed incorporation inside the hydrogel network, which hinders or controls their release from the used hydrogel base [40].

### 3.3. Physiochemical Characterizations of the Constructed Medicated Hydrogel System

#### 3.3.1. Visual Appearance, pH, and Individual Drug Content Uniformity Measurements

The constructed hydrogel was visually examined for its organoleptic characters before and after the gelation process. The results showed that all the tested formulae (n = 3) were smooth, uniform, and had no lumps or phase separation as detected in both liquid and hydrogel forms. Their colors were almost white, as all the tested samples showed the existence of the 3D structure, transparency, smoothing, and homogeneity, after they were dissolved in distilled water (the picture of the hydrogel was illustrated in supp. Info. Appendix A). From the point of that the pH value of the new formulation is very important to ensure its safety and compatibility against the targeted animal body tissues [41], in the present work, measurement of the pH value was applied on the tested number of medicated hydrogels of sample formula (n = 6). The obtained results revealed that their pH values were in the applicable range of 7.2–7.4 (Table 2), which indicates their utility for a safe nonirritant localized injection [42]. Another important issue related to the efficient loading and distribution through the hydrogel base is that the measurement and evaluation of drug uniformity, here in the developed and validated simultaneous spectrophotometric method, showed that the individual drug contents of all the tested sample solutions were used (n = 6) since all samples were in the range of 99.5% ± 0.53–105% ± 2.0%, which confirmed the accepted loading and homogeneity for further applications.

#### 3.3.2. Rheological Studies

Next, various parameters, involving the gel viscosity, storage modulus, loss modulus, and gel strength, were investigated and confirmed to give a complete insight into the behavior of the rheological properties of the proposed medicated hydrogel system. There are different factors affecting this rheological behavior including temperature, frequency, and shear stress. The results were illustrated in Table 2, showing that the constructed hydrogel system had a viscoelastic behavior, as indicated from the higher values of storage modulus (G′) compared with loss modulus G″ values.

Generally, it was reported that there is a direct relationship between the gel composition and their properties. For example, the gel system formed from higher Mw polymers has higher values of viscosity and gel strength. This may be attributed to the fact that the higher Mwt pβ-CD contains a higher number of CD cavities available for inclusion complexation. Similarly, the higher Mw of branched PEG-chol includes higher number of cholesterol moieties, which are available to form inclusion complexes with CD cavities. Consequently, the higher density of cross-linking points will be accompanied with higher gel strength and viscosity [43]. Our hydrogel system was investigated by a stress sweep experiment at a temperature of 25 °C, with 1Hz frequency. The results were illustrated in Figure 1A and Table 2. The obtained data showed that the values of storage modulus (G′) were higher than those of loss modulus (G″), confirming the formation of viscoelastic hydrogel. Both viscosity and gel strength (G*) were also investigated from the rheogram of the stress sweep. The results were 6940 Pa and 6950 Pa, respectively. Moreover, the stability of the constructed hydrogel was deduced from the stress sweep rheogram. The results showed that the modified hydrogel was stable, since it was observed that the breakdown value was 1000 (μN·m). Next, after carrying out the temperature sweep experiments on all medicated hydrogel sample solutions (Figure 1B), the results showed that the values of storage modulus (G′) decreased gradually with the increase in loss modulus values (G″), until the two panels crossed each other at a certain point (at 68 °C), called a transition temperature (T_gel_, the point at which the system transformed from hydrogel to solution). After this point, a further increase in the temperature leads G″ to be higher than G′, indicating the conversion of the hydrogel to a viscous liquid (sol), Figure 1B. Interestingly, when the sol is allowed to be re-cooled, the values of G′ will increase gradually, which was accompanied with a decrease in G″ values until crossing each other again (T_gel_), confirming the thermoreversability and the physical nature of the modified hydrogel. Values of T_gel_ were recorded and illustrated in Figure 1B and Table 2, which ranged from 68 °C. This finding matched the previously published report [44].

#### 3.3.3. In Vitro Release Studies of the Medicated Hydrogel

To confirm the efficiency of the medicated hydrogel systems for drug-mixture loading contents and to control their delivery of release, the in vitro releasing profiles of a 5-FU/MTX mixture were investigated in a PBS at 37 °C and illustrated in Figure 2A,B. Moreover, confirming the whole contents of individual drug released from the hydrogel base without any destructions or engagements of any of the individual drugs, Figure 2A showed that the release profiles of the individually 5-FU and MTX were analyzed using different samples of the medicated hydrogel (n = 3).The quantitative release of both MTX and 5-FU (in a concentration level of 0.1%, *w*/*v*) was extended for two weeks. Interestingly, it was observed that the release rate of MTX was faster than 5-FU, especially at the beginning of the release profile (40% and 60% was released within the first and second days of the release profile, respectively, compared to 15% and 38% of 5-FU released during the same period). This result may be attributed to the higher solubility of MTX (as it prepared in a lyophilized form) compared with 5-FU [45]. Consequently, another novelty can be added to the present study, with the intention of adjusting the former (faster) release of MTX, a synergist partner, before the later release of 5-FU, which appeared to possess the efficacy of the known synergistic effect of both drugs.

Regarding the effect of drug concentration and the profile of drug release, Figure 2B showed the effect of drug concentration on the 5-FU release profile (as a representative study) of the medicated hydrogel. Thus, the modified system’s results exhibited that the release rate decreased (or extended), which was indicated by the completeness of the releasing profile within 14 days, 18 days, and 21 days, with drug concentration of 0.1%, 0.3%, and 0.5%, respectively. Moreover, these results confirmed the utility of such a hydrogel system as a device for targeting and controlling the sustained release of such anticancer drugs, which was the main objective of the present work.

#### 3.3.4. Injectability Studies

To confirm that constructed gel can be injected easily, the injectability study was carried out using syringe needle system. The results showed that the volume (5 mL) of the investigated hydrogel system was extruded through the syringe needle by finger pressing within 15 s, indicating that the injection of our system is possible and easy to be achieved through syringes (i.e., no need for implantation or surgery).

### 3.4. Cell Viability Studies

The anti-proliferative effect of the prepared hydrogel system against breast cancerous cells was investigated using the MTT assay. MCF-7 cells were treated with a blank unmedicated hydrogel (negative control), 5-FU free solution, 5-FU/MTX saline solution, and the modified hydrogel loaded with 5-FU/MTX at four different concentrations (100, 200, 400, and 600 μg/mL), for the period of 24h incubation. As shown in Figure 3, the drug-loaded hydrogel as well as the drug saline solution showed a very highly significant (*p* < 0.001) observable growth inhibition ability against the MCF-7 cells, in comparison to the untreated control group. Moreover, it was observed that the multidrug treatment (5-FU and MTX) was found to have a highly significant increase (*p* < 0.01) of anti-proliferation capacity, compared to the 5-FU solution. Similarly, the multidrug-loaded hydrogels (5-FU/MTX hydrogel) showed higher inhibition of cell proliferation, compared to the single 5-FU loaded gel. These results indicated that the multi-drug co-delivery system by the modified hydrogel systems showed obvious synergistic chemotherapy effects toward breast cancer cells. Free drug application (either single or multiple) showed a highly significant (*p* ˂ 0.01) antiproliferation effect compared with the drug-loaded gels. The relatively lower effect of loaded hydrogels, compared with the free drug solution, may be attributed to the diluting effect of a gelling agent, which showed a safe nontoxic effect on the investigated cells when examined as a control. As a conclusion, all investigated medicaments, either in free form or in gel form, showed significant inhibitory profiles, with a dose-dependent manner against the investigated cancer cell lines.

### 3.5. In Vivo Studies

#### 3.5.1. Chromatographic Data Analysis and Pharmacokinetics’ Parameters

In this section, the in vivo release studies after intertumoral administration of 5-FU or combination of 5-FU/MTX, either in saline or/in the proposed hydrogel system, were carried out using female rats’ induced breast tumors. Thus, the results of measured plasma concentration against time were plotted and illustrated in Figure 4A–C. The results showed that the release of the free 5-FU solution ended within several hours (7 h) Figure 4A. Similarly, the release profile of 5-FU solution in the presence of MTX (as combination) exhibited a mild burst release at the initial stage of release, with rapid eliminations behavior as well, as shown in Figure 4B, indicating that there is no drug interaction between 5-FU and MTX when they are injected together. In contrast, the plasma concentration-time profiles of the 5-FU/MTX-loaded hydrogel showed the stable concentrations level, which reached their maximum levels after four days and steadily extended for more than two weeks, Figure 4C. These results may be attributed to the effect of our hydrogel, which allowed the sustained release of the drugs and delayed the fast elimination. Noteworthy, due to 5-FU having a narrow therapeutic index, the controlling of the serum drug concentration is very important for safety and efficacy. The blood concentration profiles for both drugs in the mixture, either in saline and/or in the hydrogel base samples, were well described by a one-compartment open pharmacokinetic model. It was observed that there were significant differences (*p* < 0.05) between the different investigated groups regarding their pharmacokinetic parameters such as C_max_ (maximum serum concentrations), T_max_, the elimination half-lives (t_1/2_), and the area under the curves (AUC). The recorded C_max_ for the 5-FU solution and 5-FU/MTX solution was 25.2 μg/mL and 26.8 μg/mL, respectively (Figure 4A,B). However, the value of C_max_ in the case of MTX was 15.2 μg/mL. This result may be due to difference in loading doses between 5-FU (100 mg/kg) and MTX (40 mg/kg). While in the case of 5-FU/MTX loaded hydrogel, C_max_ was 2.45 μg/mL (Figure 4C).

Moreover, the T_max_ for the investigated groups was 1 h for the sole administration of 5-FU and, after, was combined with free MTX solution. Whereas these values were extended to four days after the drug combined with MTX was incorporated into the hydrogel base. Considering that the circulating blood volumes for rats were suggested to equal 60 mL/kg [32], and the release of free 5-FU solution after intra administration followed a first order pharmacokinetic, it was found that the elimination half-life (t_1/2_) was 0.5 h (Kel = 138.6 h^−1^). This value was 18 h in the cases of the formulated 5-FU hydrogel, and the elimination rate constant was (n = 6) 0.014 h^−1^. Statistically, it was found that there is a significant difference (*p* = 0.017) between the groups receiving 5-FU/MTX in the proposed hydrogels and those receiving the drug mixture in saline. In addition, the AUC of any of them were nearly five-fold more than that of the drug intratumorally administered in the saline solution. These findings were attributed to the fact that the modified delivery system allowed the release of these drugs in a steady sustained manner, which prolonged the duration of their actions and decreased their clearance inside the blood of the tested rat. On the other hand, the elimination of 5-FU/MTX in a mixture incorporated hydrogel showed a significantly delayed release from systemic circulation (more than 14 days, Figure 4C, and these obtained results come in good accordance with other reported studies) [46]. Worth noting, the formulated hydrogel keeps the drug in a stable state, and there were no signs of drug degradation that have been detected during the period of in vivo investigations. The obtained data are in a good accordance with the previously reported data [32].

#### 3.5.2. Histopathological Appraisals and Traits

The microscopic examination of the mammary gland tissues of the experimental groups is displayed in Figure 5. Samples from the infected rats (G1) (untreated group, Figure 5A) showed marked histopathological changes in both mammary gland tissue and covering skin. Mammary glands acini and ducts showed enormous proliferating neoplastic cells with marked dysplasia. The neoplastic carcinoma cells formed groups and clusters with a low tendency for acini formation. In the ducts, the proliferating neoplastic cells formed multiple layers, papillary projections, or even clusters. Carcinoma cells showed the criteria of malignancy including cellular and nuclear pleomorphism, hyperchromatic nuclei, an increased nuclear to cytoplasmic ratio, and frequent atypical mitosis. Additionally, the surrounding tissues showed a marked inflammatory edema, with an intense inflammatory cell infiltration. The skin covering the affected gland showed acanthosis, with marked vacuolation in the prickle cell layer. The surrounding tissues showed a marked inflammatory edema, with intense inflammatory cell infiltration. The skin covering the affected gland showed acanthosis, with marked vacuolation in the prickle cell layer. The skin surface was covered by a thick sero-cellular crust, with intense inflammatory cell infiltration. Some of the severely affected cases showed squamous cell carcinoma in the skin covering the gland with the existence of the characteristic ‘epithelial pearl’ that was made up of central keratin whorl surrounded by neoplastic epithelial cells. Only mild improvement was noticed in G2 (5-FU saline solution, Figure 5B), the tumor cells were detected inside the ducts of the mammary glands forming inward projections or multiple layer linings. The surrounding tissue showed intense inflammatory cell infiltration.

Likewise, mammary gland tissue samples from G3 (5-FU/MTX saline solution mixture, Figure 5C) showed proliferating tumor cells in the ducts and acini of the mammary gland. Neoplastic cells were forming inward projections in the duct lumen, with the presence of inflammatory cell infiltration in the surrounding tissue. Some of the examined sections revealed some ducts and acini, with limited neoplastic proliferation and mild inflammatory reactions. Some other sections showed hemorrhagic areas, with the presence of few proliferating tumor cells within the ductal epithelium (Figure 5C). Marked improvement was noticed in G4 (Figure 5D), in which the proliferating neoplastic cells were few, with a higher tendency for duct and acini formation. No clusters or sheets of tumor cells were detected, and the inflammatory reaction was minimal. In light of the above-mentioned findings, it can be concluded that the intratumor injection of either the 5-FU or MTX solutions or the modified-drug-loaded hydrogel systems have an obvious and remarkable improvement as well as histopathological sign, compared to the untreated group. Moreover, the extent of improvement upon using both the two drugs together was better than using a single 5-FU free solution. This may be due to the synergistic effect of MTX. The loaded-drug hydrogel systems showed a higher improvement pattern, compared with the free-injected drugs. The results might be attributed to the effects of hydrogel, which keep the drug inside the cancer tissues for a relatively extended time, compared with the free drug, which consequently causes more curing of the cancer. The best results were obtained in the case of the hydrogel system G6, which showed the highest delay of drug release, as discussed in the HPLC section.

#### 3.5.3. Effect of the Proposed Hydrogel System on the Tumor Growth

Regarding the change in a tumor volume, which was monitored after drug administration, the results illustrated in Figure 6 showed the result of applications of different samples (n = 6), of 5-FU sole, and/or with MTX in a mixture that was administered in normal saline, as well as a 5-FU/MTX mixture in the proposed hydrogel. The results were gathered after 5-, 10-, and 15-day intervals.

The data were significantly progressed in order of decreasing the tumor growth rate and achieving a significant increase (*p* = 0.001) in the antitumor efficacy of rats, as compared to the controlled group samples, all from the periods of the study. Additionally, it was found that there is a significant difference (*p* = 0.0017) between whether the drug mixture was administrated in a saline vehicle or in the proposed hydrogel-controlled system, for all the investigated groups’ samples (n = 6), in comparison with each other. The pictures of the breast-cancer-induced rats before and after treatment were illustrated in Appendix A.

#### 3.5.4. Tumor Markers Detection in the Rats’ Blood

The serological examination, of the induced cancer markers in the rats’ blood, was performed depending on the quantitative determination of both CA and CEA. A CA marker was measured to realize the presence of breast cancer, while CEA indicated the progression of this cancer to other rat tissue or organs. Figure 7 listed the recorded data of both two tumor markers (CA and CEA), for all tested grouped members (n = 6) after two weeks of treatment. The results indicated that both the investigated parameters are high in G1 (the controlled group; cancer-bearing rats), which showed a higher level (12 folds) of tumor markers when compared with the normal rats, while the values of these two tumor parameters were reduced significantly (*p* < 0.05) in all the treated grouped rats’ samples. Additionally, the levels of TNF-α and IL-6 were elevated in the control group, which received no medication, while their levels were reduced significantly (*p* < 0.05) after injection of the modified hydrogel loaded with antitumor mixture (G4). These obtained data are in a good agreement with the previously obtained histopathological data, which showed the reduction in swelling and inflammation accompanying the tumor proliferation.

## 4. Conclusions

A straightforward procedure was established for the formulation of a stable hydrogel system, after a simple and inclusion complexation process between a pβ-CD and a branched PEG end-capped with a hydrophobic moiety of cholesterol. All the used polymers were synthesized and characterized carefully by the authors. A rapid rehydration process with PBS was utilized in the incorporation and careful distribution of a 5-FUµl/MTX mixture into the prepared hydrogel base. Throughout the present study, the in vitro release results and in vivo studies exhibited the utility of the constructed hydrogel system for loading and controlling the sustained release behavior of both the two investigated chemotherapies for more than four weeks. Collectively, the above-mentioned finding confirmed the impact and the novelty of the new assembling, medicated, in-situ-injectable, and controlled-release hydrogels as an efficient and biocompatible device for the loading and release of the important chemotherapy used for the treatment of breast cancer.

## Figures and Tables

**Figure 1 pharmaceutics-14-00817-f001:**
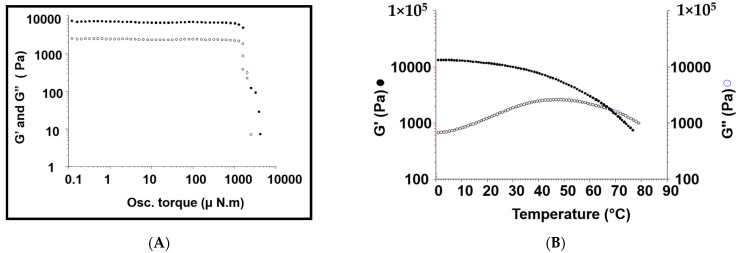
(**A**,**B**) Storage modulus (G′, solid shapes) and loss modulus (G″, empty shapes) of the hydrogel system samples (n = 3), deduced from both oscillatory stress sweep experiments at 25 °C, 10 (μN·m), and 1 Hz frequency, (rheogram (**A**)); and a temperature sweep experiment, carried out at temperature rate of 1 degree/min and 1 Hz frequency, (rheogram (**B**)).

**Figure 2 pharmaceutics-14-00817-f002:**
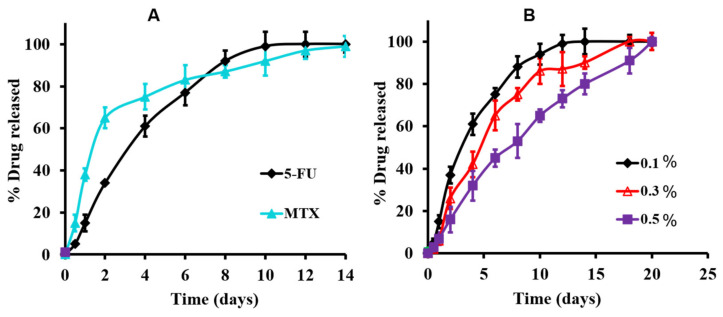
In vitro release profiles of the individual 5-FU and MTX from the modified hydrogel at 37 °C in PBS at 0.1% drug concentration (**A**), and the release profile of 5-FU as a function of drug concentration in PBS at 37 °C (**B**).

**Figure 3 pharmaceutics-14-00817-f003:**
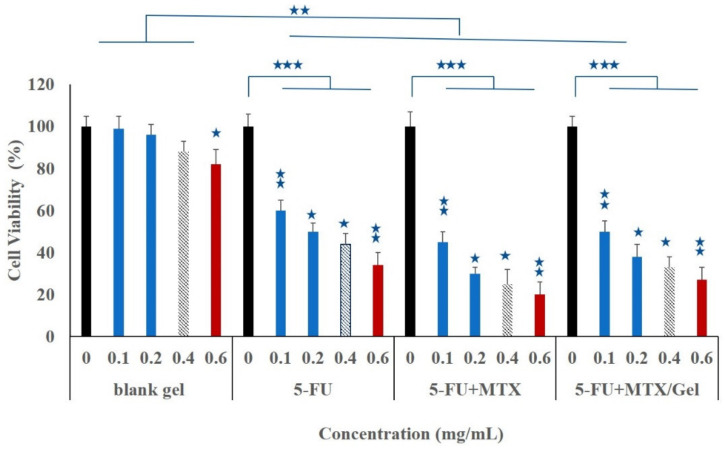
Cytotoxicity studies of 5-FU-loaded hydrogel, as a function of concentration, on breast cancer MCF-7 cell membranes. The results were illustrated in comparison with either unmedicated hydrogel (blank gel) or free drugs solution after incubation with MCF-7 cell for 24 h. The results are shown as mean (n = 3) ± standard deviation. ⋆ *p* < 0.05, ⋆⋆ *p* < 0.01, and ⋆⋆⋆ *p* < 0.001 for significant, highly significant, and very highly significant, respectively. Unpaired Student’s *t*-test.

**Figure 4 pharmaceutics-14-00817-f004:**
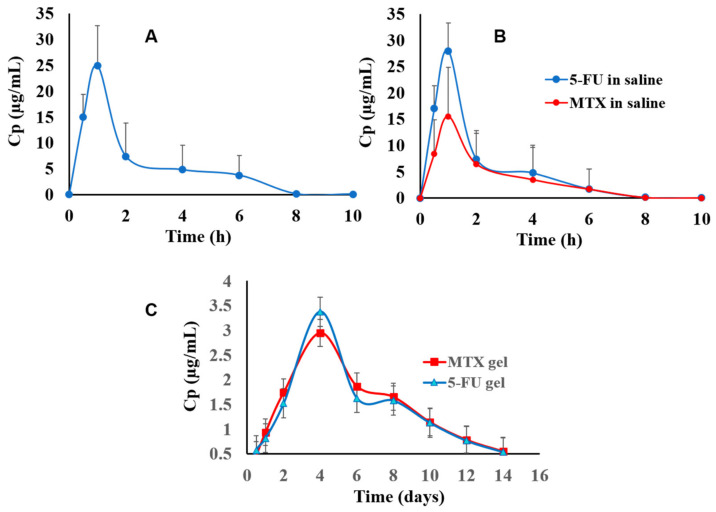
(**A**–**C**). In vivo release of intra injections of sole administrations of 5-FU (100 mg/kg, dose in saline) (**A**); injections of the combined administrations of 5-FU/MTX (respectively, 100 and 40 mg/kg, dose in saline) (**B**); injections of the combined administrations of 5-FU/MTX (the same dose in the hydrogel system) (**C**).

**Figure 5 pharmaceutics-14-00817-f005:**
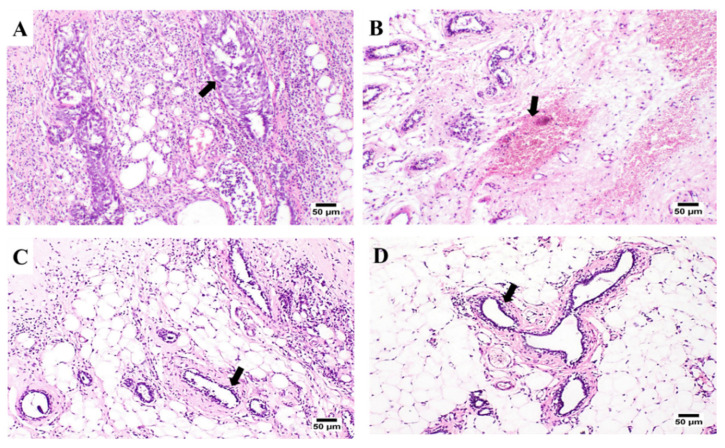
(**A**–**D**): Photomicrographs, showing some histopathological appraisals detected in the mammary gland tissues from six treatment groups (arrows). (**A**) G1 sample showing intense inflammatory cell infiltration with the existence of proliferating neoplastic cells in the mammary ducts, (**B**) G2 sample showing hemorrhage and edema in the tissue surrounding the mammary tissue, (**C**) G3 sample showing few proliferating mammary ducts and inflammatory cells infiltration, and (**D**) G4 sample showing few neoplastic cells forming ducts with mild inflammatory edema, after being treated with the proposed hydrogel incorporating the drug mixture. The fixed samples were processed for the usual histological procedures of the hematoxylin and eosin 100× (H and E) counter staining of ultra-thin paraffin sections (5 µm; Micro HM 360^®^ Microtome).

**Figure 6 pharmaceutics-14-00817-f006:**
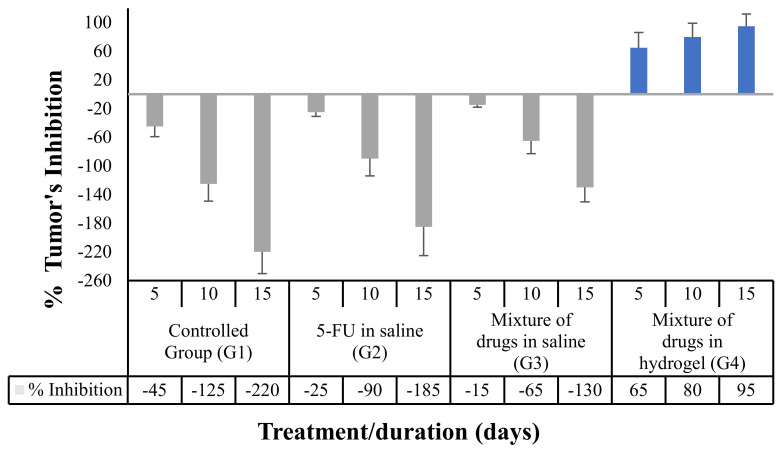
The effect of an application of medicaments in different vehicles on the behavior of the tumor’s inhibition %; the effects of applications were evaluated after 5-, 10-, and 15-day intervals after treatments; where: G1 is the controlled group; G2 is an individual 5-FU administered in normal saline solution; G3 is a mixture of 5-FU/MTX solution administered in saline; and G4 is the proposed medicated hydrogel formula. The results are presented as the mean of three independent experiments ± SD (n = 6).

**Figure 7 pharmaceutics-14-00817-f007:**
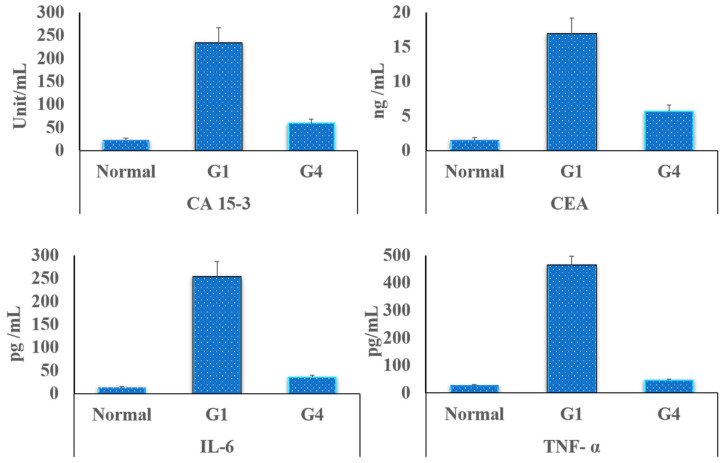
The tumor markers measured in the grouped rats’ blood after two weeks of the applied treatment, with the proposed hydrogel incorporated 5-FU/MTX mixture. G1 is the untreated controlled group, and G4 is the treated group (n = 6). CA 13-5 is the cancer antigen 13-5; CEA is the carcinoembryonic antigen; TNF-alpha is the tumor necrosis factor-alpha; and IL-6 is the interleukin 6.

**Table 1 pharmaceutics-14-00817-t001:** The degree of substitution, molecular weights, and yield of the modified polymers utilized for hydrogel base construction.

Polymer	Degree of Substitution (%)	Molecular Weight (kDa)	The Technique(s) Used for Mw Determination	Yield (%)
pβ-CD	62	106	DLS	62–65
8armPEG20-(Chol)7	87	22.7	NMR, MALDI	85–90

**Table 2 pharmaceutics-14-00817-t002:** The physicochemical and rheological properties of hydrogel samples (n = 3).

Drug Content		pH	T_gel_ (°C)	Cross-Over (Hz)	Gel Stability (μN·m)	Viscosity (Pa)	G* (Pa)
5-FU	MTX						
105 ± 2.0	99.5 ± 0.5	7.4 ± 0.1	68	0.1 ± 0.03	1258 ± 30.2	6940 ± 70.2	6950 ± 84

G* means gel strength.

## Data Availability

All data are contained within the article.

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
