# Peer review of "Polymerized β-Cyclodextrin-Based Injectable Hydrogel for Sustained Release of 5-Fluorouracil/Methotrexate Mixture in Breast Cancer Management: In Vitro and In Vivo Analytical Validations"

_pharmaceutics, 2022, doi:10.3390/pharmaceutics14040817_

Round 1
Reviewer 1 Report
This manuscript presents the synthesis and biological evaluation of hydrogel for dual and sustained release of two anticancer drugs. Overall, I feel it would fit the scope of this journal. However, some concerns need to be clarified before that.
- English should be checked by a native speaker.
- Line 120. How do you determine molecular weight from DLS? At some point it is mentioned that some dynamic size correlates with a molecular weight in particular. Where does that come from?
- Line 136. I wouldn’t say that a 5M HCl solution is a diluted solution.
- Line 141. Why deuterated water instead of regular water?
- Line 148. What is 3D masses?
- Regarding the release of the drugs, is it possible that the drugs degrade throughout the 14 days in an in vivo situation?
- Section 2.5. Indicate wavelengths at which drugs are detected.
- Section 3.3.3. I would present the release experiment of the final material. That with the chosen concentration of both drugs. Why those concentrations were chosen? Based on what?
- In vivo experiments. What the physical state of the gel when injected? A picture could clarify this. At some point it is mentioned that above a certain temperature (>60ºC) the gel becomes a viscous liquid. Then, how is the hydrogel injected?
- Line 307-309. That sentence is very hard to follow.
- Could it be possible to provide a picture of the hydrogel on the tumor?
Reviewer 2 Report
The manuscript of Almawash et al. presents a potential application of beta-cyclodextrin-based hydrogel. The current version needs a deep revision and completion because the experimental part suffers from serious weaknesses. The soluble ßCD polymer is inappropriate for parenteral applications - intratumoral use is considered parenteral administration. Additionally, the authors failed to properly characterize the materials used, which calls into question the reproducibility. It is hard to believe that the description suggested poor synthetic knowledge can result in "good science" in the biological section. The reference section is inconsistent because many references do not contain the DOI, despite the availability.
Without completeness, some points are below.
- 50% of the Keywords are from the title. The authors even did not care too much about the order of words. Do the authors explain which additional information is in the redundant words?
- Source of FU and MTX is missing.
- Reference 19: there are only a few connections between the adamantane-based acrylamide and ßCD polymer and cholesterol-based hydrogels. Additionally, not Koopmans&Ritter reported first the synthesis of epichlorohydrin/CD (soluble) polymer, and this can be true for the PEG-cholesterol system, too.
- The description of the CDPs is incomplete. At first, 5 ml of epichlorohydrin is NOT 5 millimole (line 113). The unreasonable "extraction" (lines 114-115) - what is extracted from what - and dissolving the precipitated crude product in deuterated(!) water (line 116) are also strange.
The DLS is not spectroscopy but a technique (line 121). The evaluation of the autocorrelation function is called photon correlation spectroscopy.
- In lines 140-142, the dissolution and immediate freeze-drying of the two hydrogel components and then dissolution in PBS is not only unreasonable but meaningless, and the description of the pH measurements is a complete mess. What prevented the authors from dissolving the components in PBS directly?
- In line 238, the equation contains some errors, perhaps did the authors want to write (exp(-kel/t)-exp(-ka*)?
In the (Ka F D), the mathematical operators are missing.
Is Ka equal to ka? If so, why did the authors use two different identifiers?
In addition, because the Cp is an explicit function, knowing the parameters, the numerical integration seems inappropriate.
- Line 292, the used statistical software needs a reference to its homepage.
- Lines 310-312 contains MW estimates from the hydrodynamic diameter of the DLS experiments. It can only be correct if the authors used an appropriate calibration material. Due to the missing calibration polymer and the undefined shape of the CD polymer, the 96kDa value appears to be incorrect. On the other hand, the authors used ~60 millimoles of cross-linker, which, assuming that all epichlorohydrin reacts with all CD, at best the MW cannot be greater than ~72kDa (~60*(1135+56) /CDO-CH2CH(OH)CH2-OCD/). The epichlorohydrin is a very reactive molecule, i.e., in a basic solution, its hydrolysis inevitably occurs, so not all CDs will be linked at the end of the reaction. It also means that the polymer contains (2,3-dihydroxy)propyl side chains that can only react with epichlorohydrin and not with other CD molecules. This side reaction further reduces the MW. Reference 20 reports on a completely different polymer.
Unit mM is a concentration unit.
- NMR experiments are mentioned, but instrument and experimental conditions are unknown. The MALDI is sensitive for the matrix, particularly in the case of CDs, and the experimental parameters are also unknown.
Table 1 contains a few invalid pieces of information. NMR is unsuitable for the MW calculation, and in case of lacking internal standard, only the calculation of CD: linker ratio is possible. According to the text, the authors performed both NMR and MALDI experiments, so they need to provide the spectra and assignments in SI. Seeing is believing.
Does the degree of substitution mean CD or cholesterol content? In this viewpoint, the degree of substitution is difficult to interpret. Principally, it is a ratio (number substituents/CD) and therefore has no dimensions (%). Or, in the CD polymer case, DS means the CD hydroxyl substitution?
- What does "the higher density of the cross-linking points " mean in line 357? Does it characterize the compactness of a CD polymer? How is this parameter calculated?
- Although Section 3.3.2 is descriptive, discussion of the extremity (loss modulus) of Figure 1B is missing. The curve can have another extremity (minimum) near 0 oC.
- It is not clear why the higher aqueous solubility of MTX explains its slower release from the hydrogel. It can be advantageous also to use another color for 0.3% Fu content in Figure 2B because in Figure 2A, red means MTX release.
- Is there an explanation for the two Cp maxima in Figures 3A and 3C?
- Staining information is missing in the images in Figure 4.
- In line 560, the authors stated that all data are in the text. The statement tangentially touches the truth only. Both the NMR and MALDI spectra and the particle size distributions are missing.
- DOI is missing in references 16, 19, 21, 35, 44.
In reference 11, the web page and last accessing time are missing.
In reference 17, the journal name is unabbreviated.
Reference 42 seems incomplete.
Reviewer 3 Report
The manuscript describes the results on polymerized β-cyclodextrin-based injectable hydrogel developmed for sustained release of 5-fluorouracil/methotrexate mixture in treatment of breast cancer. The subject is interesting, but the manuscript has serious flaws and needs serious improvement before considering for publication in Pharmaceutics.
The major points:
- The novelty of the study has not been explained.
- Characteristization of the hydrogel is very limited. First of all, the binding of the drugs with cyclodextrin has not been proved and characterization of the complexation process has been completely ignored. Moreover, according to the literature the binding of MTX with polymeric b-CD is not sufficient compared to the b-CD and dimeric b-CD, probably due to the steric hindrance, which was not discussed in the manuscript.
- The NMR and FTIR analysis would be beneficial to chracterize the developed drug carrier
- The cytotoxicty of the hydroges has not been evaluated, that would prove that the anticancer effect is caused by the drug. No in vitro study on normal and cancer cell lines has been conducted before in vivo analysis, which should be a golden standard. Moreover, drug-free hydrogel has not been used as a control in animal study to compare this effect with drugs-loaded hydrogel
Minor points:
- citation [23] appears only in the references list at the end of the publication, but it is missing in the text
- there are numerous editorial errors that should be corrected
Reviewer 4 Report
The draft on inclusion complexes of polymerized β-CD/PEG-Cho-based injectable hydrogel and their analytical validation of two drugs (5-FU/MTX) used in cancer treatment. The in vivo results on mice are very encouraging, although the translation always has the complexity derived from scaling up and clinical trials.
As for the content of the paper, it is well developed, well understood and well explained. Some typographical errors that should be corrected. Finally, the work casts a series of general doubts together with other specific ones that should be supported.
The preparation and characterization of the hydrogel is based on previous articles (19,20), in which hydrogels with slightly different molecular weights and properties are obtained. However, these results are not described in this work, which is expressed in a simple table without endorsement.
- The authors are requested to include a support of these appended data in supplementary material. Resonance spectra and calculations to obtain the degree of substitution and molecular weight should be included.
- Why are there yield ranges? Are there any bathes?
- The batches of each polymer may have influence on the final properties, either rheological, or drug encapsulation or release and all this goes unnoticed by the authors. How to validate a complex that may present batch-to-batch variability in the synthesis of the hydrogel?
- They should revise the numbering of the tables, both in the headings and in the text.
- In table 2 (currently table 3, on page 8), the expression in the precision of the results is not correct; its format and formalism should be improved.
General question: How do the molecular weights of each complex affect the results?
In the in vitro vs in vivo analysis, two different analysis techniques are used. Why has chromatography not been used in the release drugs?
In my opinion, the spectroscopic part will be validated. Comparison between the two assays would improve the perception of validating this methodology.
Round 2
Reviewer 1 Report
Authors have answered my questions
Author Response
Thank you very much for your valuable comments and your efforts
Reviewer 2 Report
The authors answered the majority of concerns, and they are mostly correct, but the MW of ßCD polymer is still debated and needs further confirmation. The two points below can be corrected by a major revision only.
- Answer to point 4, handles the truth with easy hands. The original version contained more than 50% of copied/pasted words from the title. In the current version, the authors have neither replaced nor reduced the number of repetitions. Please remove all or the majority of the following words: cancer, hydrogel, polymer, 5-Fluorouracil, methotrexate. These are in the title and do not provide additional information or indexing benefits.
- Answer to point 13 is confusing.
a) Finally, the epichlorohydrin was neither 5 ml nor 5 g but 4.62 g (0.050 mol). At least it is in the text.
b) 10 g native ßCD contains 14% water. It is unclear whether the authors used dried or hydrated ßCD (11.6 g). Although this is an essential but less cardinal issue, it does affect yield and MW.
c) According to the authors, they used 8.82 mmol ßCD and five mol% (for what? CD?) epichlorohydrin. The 5 mol% for ßCD is 0.44 mmol (0.041 g). If we count on OH groups (in the case of ßCD, there are 21 substitutable OH), there are ~185 mmol (21*8.82) OH, i.e., 50 mmol epichlorohydrin is 27 mol%.
d) The authors claim to have produced a "linear" ßCD polymer, which may be valid but is hardly plausible because, without protecting groups, the large number of OH groups makes it impossible to form only two cross-linkers/CD.
e) Although the authors have modified the MW in Table 1, NMR is still in its calculation method. Again: the MW cannot be calculated from the NMR spectrum if the authors did not use an internal standard (e.g., HPßCD). From the spectrum, we can calculate only the ßCD content. Set the integral of anomeric proton =7, and the number of glycerin residue NG=(other than anomeric protons integral - 42)/5. The reciprocal value*100 of NG shows CD% (moles), the weight% is CD%=1135/(1135+NG*58)*100. In this case, this value is ~70% from the provided NMR in SI.
The assignment of 7/9 protons near 2.8 ppm is incorrect. Those signals more of the epichlorohydrin end (the opposite CH2 to the chloromethyl).
f) The MW is still under dispute, and the referee keeps the original opinion: that value is less probable if the assumed polymer is linear (CD-linker-CD-linker-...).
Back calculation from their data, if the yield is 65% (~8 g product, assuming no (2.3-dihydroxy)propyl groups), and the CD content is 62%, then in the product, the CD/glycerine skeleton ratio is near 1:12 (CD=8*.0.62g, linker=8*0.38g => 5g CD=4.4 mmol, 3 g linker=52 mmol, 52/44=11.8, which is the maximum number because the presence of (2.3-dihydroxy)propyl group decreases this value). It means the polymer is not only branched but has many 2,3-dihydroxy)propyl sidechains. The linear version should have higher ßCD content.
Under the applied synthetic conditions, the epoxy utilization is around 60-67% (strongly depends on the NaOH concentration and temperature), which means the crosslinking epichlorohydrin is <40%, and the MW >100K seems unreal. Or the yield is not correct, but the authors missed including the weight of the purified polymer. Please include this value in the appropriate section.
Regarding the molecular shapes, the assumed linear version surely needs a similar calibration substance for the correct calculation in the DLS experiments.
The authors have MALDI, then why did they not use that for the ßCD polymer?
Author Response
Thanks a lot for your effort and the very valuable comments. Accordingly, the manuscript has been improved and our response is attached.

Reviewer 3 Report
The revised version of the manuscript is significantly improved, which increased its scientific value. However, in my opinion a further revision is needed before considering the manuscript for publication:
- Section 2.7, line 225: "Then the culture medium was replaced with 500 μL" - please, check the added volume of medium since it seems too large for 96-well plates.
- Section 2.7. Please, add the sterilization method of the studied hydrogels.
- Figure 3 - please, mark the statisticaly significant differences with asteriks
- Section 3.4. "The anti-proliferative effect of the prepared hydrogel system against breast cancerous and normal cells was investigated using the MTT assay" - this sentence should be corrected, because the normal cell line was not involved in the study.
- Section 2.7 and 3.4 - To show the cytocompatibility of the drug-free material, it is necessary to use cells cultured in medium (unmodified culture, without any material) as negative control. Otherwise, the exact influence of the drug-free hydrogel is still unknown.
- Section 3.4 line 493: "for the period of 24h incubation" - according to the Section 2.7 the incubation time of the MCF7 cells in the presence of the studied materials was 48 h - please verify the incubation time.
Author Response
Thanks a lot for your effort and very valuable comments. Accordingly, the manuscript has been revised (please see the attached)

Reviewer 4 Report
Your manuscript has become more consistent and understandable.
Author Response
Thanks a lot for your effort and very valuable comments. The whole manuscript has been revised accordingly.
Round 3
Reviewer 2 Report
Although the manuscript has been further improved again, some inconsistencies have remained. The synthetic description is out of the standards, and the referee cannot accept the authors' explanation. They do not know the structure of the polymer.
- One of the most important, and still very far from the standard chemical terms, is the use of '5 mol%'. The whole explanation shows very distorted thinking, and additionally, it is not consistent. According to the authors, the 1/100 of 5 molar fold EP (0.050 mol, or 50 mmol) of EP is 5mol%. OK, but is there a reason for calculating the CD amount differently? Why did they write 8.82 mmol and not 8.82mol%? The referee does not accept this unconventional calculation which is an incorrect reinterpretation of the synthetic chemistry units. 4.42 g EP is 50 mmol, while 10 g ßCD (dried) is 8.82 mmol. By the way, what does "gm" mean? Is it an archaic version of the gram? The authors know the correct version as they could use it correctly elsewhere.
Also, the authors' recommendation to read the publications of Yang and Koopman is very kind. Unfortunately, the reviewer has already read them and formed an opinion on their content. But what does this have to do with non-standard synthetic description? The content of those papers is irrelevant to this manuscript. Presumably, if the current reviewer had been the reviewer of those papers, this deficiency would have been corrected in time. Authors need to understand that there are publishing rules and that they must abide by them. The 50 mmol is 0.050 mol and not 5mol%, whatever others had written!
The authors have accepted the "linear polymer" concept without verification and now point to others, though they do not know what material was in their studies. As written previously, five molar fold epichlorohydrin unavoidably reacts with more than one-to-one hydroxyls of the CDs, which means the branched polymer version is more probable than a linear one. Using a linear polymer to calibrate the DLS MW instead of a branched polymer is incorrect. Or conversely, assuming that the authors have made a linear polymer, something globular in the calibration gives unreliable results.
- Again, the /CD-/O-CH2CH(OH)CH2-O/-CD/ signals are not appearing between 2.6-2.9 ppm (line 357). Those signals appear in the 3.1-4.1 ppm range, while the signals between 2.6-2.9 ppm belong to unreacted epichlorohydrin (ring CH2) and not the glycerol moieties!
- The unit 'μ.N.torque' is not SI (line 191 and Table 2 heading), but what is even more important, what are the points inside? The correct unit can be either μNm or, because the value is in the thousands range, probably the mNm. In some scientific papers, a middle point is necessary to distinguish the various units (depends on journal requirements), the μN・m or mN・m might be acceptable, but the authors' version is out of this either.
- In both previous reviews, the referee mentioned that NMR is unsuitable for MW determination, but the recent text still contains the NMR as a tool in the MW calculation. In the feedback, the authors themselves wrote that they did not use NMR for the MW calculation, but it is still in Table 1.
- In line 498, the authors wrote 5-FU®. If they feel the necessity of using the 'registered' symbol (®) once, they should use it whenever 5-FU is in the text, or if it is unnecessary, they should not use it at all.
Author Response
thanks a lot for your effort and valuable comments. please find the attachment, including our response.
best regards

Reviewer 3 Report
The manuscript has been revised according to the comments. Therefore, I recommend it for publication.
Author Response
thanks a lot for your efforts and valuable comments
Round 4
Reviewer 2 Report
The authors modified the manuscript according to the recommendations, and it is now ready for publication.